# DanceCamAnimator: Keyframe-Based Controllable 3D Dance Camera Synthesis

## ABSTRACT

Synthesizing camera movements from music and dance is highly challenging due to the contradicting requirements and complexities of dance cinematography. Unlike human movements, which are always continuous, dance camera movements involve both continuous sequences of variable lengths and sudden drastic changes to simulate the switching of multiple cameras. However, in previous works, every camera frame is equally treated and this causes jittering and unavoidable smoothing in post-processing. To solve these problems, we propose to integrate animator dance cinematography knowledge by formulating this task as a three-stage process: keyframe detection, keyframe synthesis, and tween function prediction. Following this formulation, we design a novel end-to-end dance camera synthesis framework **DanceCamAnimator**, which imitates human animation procedures and shows powerful keyframe-based controllability with variable lengths. Extensive experiments on the DCM dataset [51] demonstrate that our method surpasses previous baselines quantitatively and qualitatively. We will make our code publicly available to promote future research.

## CCS CONCEPTS

• **Applied computing → Media arts**.

## KEYWORDS

Dance Camera Synthesis, Dance Cinematography, Keyframing

## 1 INTRODUCTION

The role of camera work in dance performances is crucial as it significantly influences how the audience perceives and understands the dance piece. By involving multiple camera switches, the producer can capture the subtle movements and facial expressions of dancers to showcase more dance details. Moreover, creative camera techniques like quick cuts, slow motion, and dolly shots, among others, can deliver visual impact and novelty to the audience, increasing the attractiveness of the dance performance. As a result, automatically producing camera movements from music and dance is an appealing but challenging task since camera movements are composed of variable-length continuous sequences and dramatic changes between these sequences, which respectively denote the camera shots and camera switches, as shown in the left of Figure 2.

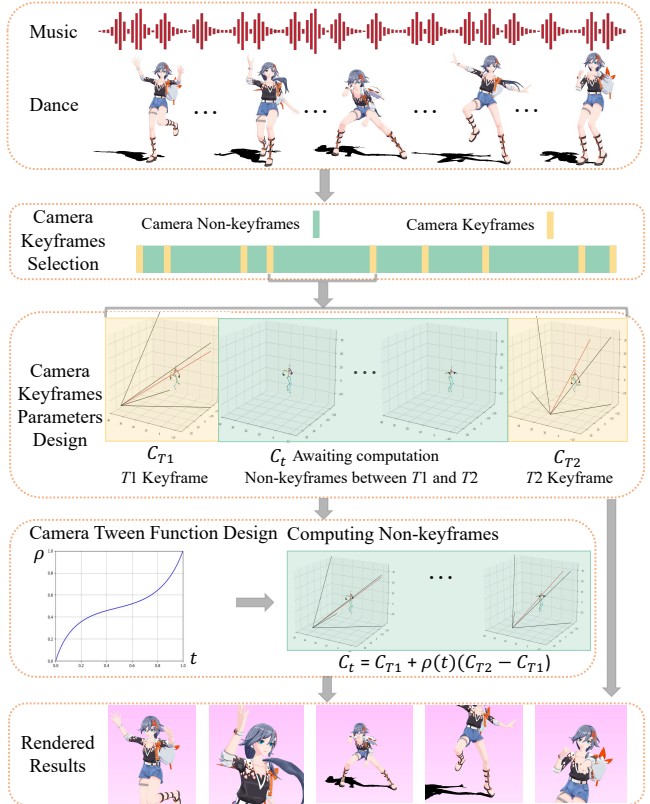

**Figure 1: Hierarchical dance-camera-making procedure by animators. According to the given music and dance, animators first select keyframes on the timeline. Next, animators set the camera parameters at each keyframe to capture the dance details or highlights. Then, for the non-keyframes between keyframes, animators produce the camera movements by editing tween curves that control the camera moving speed from one keyframe to the next. Finally, the 3D engine can render results with camera movements and dance.**

DanceCamera3D [51] constructed the first 3D dance-camera-music dataset DCM and has shown the rationality of music-dance-driven camera movement synthesis. However, it treats all frames equally and ignores the sudden changes between camera shots. This greatly affects the model's generative ability because the model cannot determine whether to generate continuous or abrupt movements resulting in jittering sequences and shakes in the final dance video, as shown in the right of Figure 2. Thus, DanceCamera3D has to conduct a smoothing post-processing by detecting keyframes using a total variation denoiser in camera parameters and filtering the frames in between. However, this post-smoothing may misjudge some keyframes and introduce some erroneous smoothing causing the camera to lose focus on the dancer. In addition, Xie

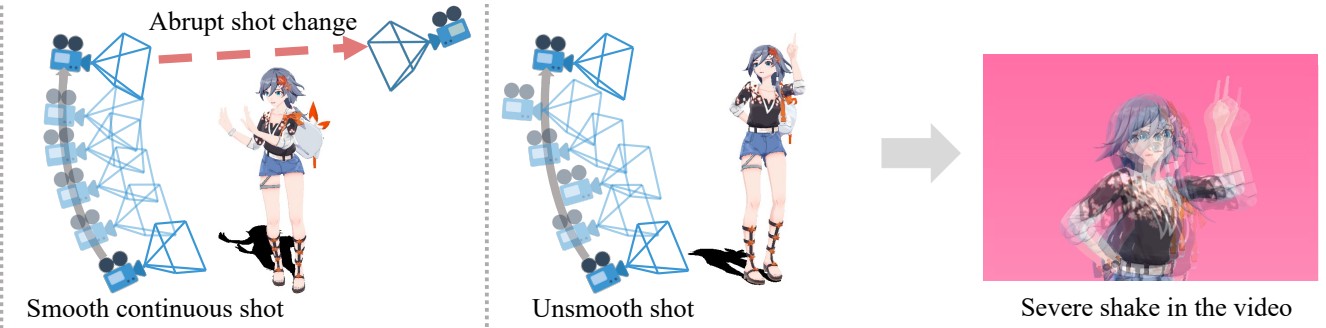

**Figure 2: Challenges in 3D dance camera synthesis. Dance camera movements are not entirely continuous because they consist of smooth complete shots and abrupt shot changes. Moreover, small disturbances can lead to big shakes of the dancer in the rendered video. These issues prevent neural networks from synthesizing satisfactory dance camera movements.**

et al. [55] proposed to generate camera movements based on the performer's position, but they ignored the changes in the camera's field of view and also simplified the question from 3D to 2D by excluding the camera orientation in terms of roll and pitch, which greatly reduces the expressiveness of camera movements and the complexity of the problem. To make matters worse, this method needs keyframe positions on the timeline, further diminishing the model's usability and automation capacity. Apart from the above two works, significant efforts have also been devoted to camera planning and control [13, 16, 20–22, 25, 26, 42, 43, 54, 58]. These works all focus on gaming and movie scenes but neglect dance camera synthesis, which is a more complex problem since it is influenced by various factors including music and dance. Due to the aforementioned reasons, although significant progress has been made in camera planning and control, dance camera synthesis remains an underexplored and challenging task.

Similar to music-dance-driven camera movement synthesis, previous works [2, 5, 6, 11, 14, 23, 29–36, 40, 44, 46–48, 50, 52, 53, 56, 57, 59] have made progress in music-dance synthesis. However, compared to human dance motions, dance camera movements possess higher complexities, highlighted by the fact that dance camera movements are dancer-centric, not purely continuous, and sensitive to jittering as shown in Figure 2.

To address the above challenges, we propose to integrate human animation knowledge into the problem of music-dance-driven camera synthesis. As shown in Figure 1, in the actual process of creating dance camera movements, animators first select keyframes on the timeline, then determine the camera parameters of keyframes, and finally modify the tween curves which are used to control the changing speed of the camera parameters from one keyframe to the next. After observation, we find that the tween curves are monotonically increasing so that the smoothness of complete shots can be guaranteed. For example, in the MikuMikuDance [1] engine, the producers provide monotonically increasing Bezier Curves for animators to edit the in-between camera movements. Utilizing this knowledge, we devise **DanceCamAnimator**, a three-stage controllable framework that synthesizes 3D camera movements from music and dance following the hierarchical camera-making procedure of animators. As shown in Figure 3, our DanceCamAnimator consists of a Camera Keyframe Detection model, a Camera Keyframe Synthesis model,

and a Tween Function Prediction model: the Camera Keyframe Detection model distinguishes whether each frame is a keyframe according to music and dance; the Camera Keyframe Synthesis model infers camera parameters at keyframes from the history of camera movements and the music-dance context; the Tween Function Prediction model learns the mapping from music-dance context, camera movement history and keyframes camera poses to the tween function values for the calculation of in-between camera movements. In this manner, our DanceCamAnimator can better comprehend a complete shot and switch between shots. Moreover, our unique design provides DanceCamAnimator with keyframe-level controllability through adjustments to the temporal positions and camera parameters of keyframes. To overcome the jittering of the camera, we generate tween function values instead of camera parameters in the Tween Function Prediction model so that the camera will move from one keyframe to the next at different speeds without moving in other directions.

To demonstrate the efficacy of our method, we conduct extensive experiments on the standard benchmark DCM [51]. Both qualitative and quantitative evaluations show that our method outperforms baseline methods. In summary, our contributions are the following:

- We propose to incorporate expertise from the animation industry into the 3D music-dance-driven camera synthesis framework by integrating and formalizing the dance camera-making procedure of animators.
- We imitate the camera-making procedure of animators to devise a novel end-to-end three-stage dance camera synthesis framework **DanceCamAnimator** that combines state-of-the-art performance with keyframe-level controllability.
- We propose to predict tween function values between keyframes instead of directly predicting the dance camera movements, thus achieving smoother camera curves and more stable dance camera shots and significantly improving the user viewing experience.

## 2 RELATED WORKS

### 2.1 Camera Planning and Control

Designing camera movements is a challenging task due to the inherent complexity of camera movements and various influence factors according to the actual circumstances. Thus, many researchers

have made attempts to synthesize or control camera movements. Early works formulate the camera planning problem as a constraint-satisfaction problem [10] and use constraint-based optimization approaches [3, 7–9, 41] to solve it. With the development of neural networks, deep learning-based models have become the mainstream solution for addressing camera auto-generation issues. Jiang et al. [24, 25, 26] constructed a dataset with film clips, corresponding camera movements, and motions of actors. This dataset facilitates their research on synthesizing camera movements from reference film clips or text descriptions with LSTM [19] and diffusion [18] models. Wu et al. [54] propose to synthesize camera movements in the storytelling scenes based on a GAN-based controller. To reproduce films in the 3D virtual environment and manipulate camera movements, Jiang et al. [27] build a differentiable pipeline to estimate and optimize the camera movements and human motion from film video and retarget them to camera and avatars in the 3D engine. Considering the requirements of the gaming scenario, Li and Cheng [37] developed a camera control module to track the player in a third-person perspective automatically, Rucks and Katzakis [43] devise CameraAI which can avoid occlusion when chasing the player, and Evin et al. [13] integrate the cinematography knowledge to develop a semi-automated cinematography toolset Cine-AI for generating in-game cutscenes. Additionally, some other works [15, 16, 20–22] have investigated the auto-driving of camera drones for filming dynamic targets using the experience of film-maker or artist. Compared to camera synthesis in game or film scenes, dance camera auto-generation is a more complicated problem due to the multifaceted influence factors including shot type changes and correlation between music, dance, and camera movements. Xie et al. [55] have tried to generate camera movements from the poses of the dance performer, but ignored the influence of music and their model needs extra input of devised keyframes on the timeline. To solve this problem, Wang et al. [51] construct the first 3D dance-camera-music dataset DCM and present a transformer-based diffusion model DanceCamera3D to solve the dance camera synthesis problem. However, Wang et al. [51] overlooked the co-existence of smooth continuous shots and abrupt shot changes in dance photography. As a result, DanceCamera3D needs additional smoothing post-processing but it's hard to strike a balance between increasing the shot smoothness and maintaining camera switches. Moreover, Jiang et al. [24, 25, 26], Xie et al. [55] considered directly synthesizing camera parameters between keyframes using neural networks, but these solutions also produce inevitable jittery camera movements and unsatisfying shaky videos to the audience, so that previous researchers have to use post-smoothing filters or use a simplified 2D camera representation which greatly reduces the diversity of camera movements and shots. For the aforementioned reasons, 3D dance camera synthesis remains a challenging problem.

## 2.2 Music to Dance Synthesis

Music-to-dance problem shares many commonalities with dance camera synthesis including the processing of music audio and dance motion, as well as spatio-temporal feature extraction. Early works [4, 5, 14, 32, 38, 40] formulate music-to-dance as a similarity-based retrieval problem that greatly limits the capacity and diversity of generated dance. Recently, with the significant advancements in deep learning models, various neural networks have been applied to music-to-dance synthesis. Initially, Crnkovic-Friis et al. [11], Li et al. [34], Tang et al. [48] tried to synthesize dance motions frame-by-frame with sequence-to-sequence models including RNN, LSTM, and Transformer. Besides, Kim et al. [28], Wu et al. [53] employ Generative Adversarial Network (GAN) to investigate more generative capabilities including genre control and dual learning between dance and music. More recently, Chen et al. [6], Ye et al. [57] synthesize pre-annotated dance units instead of dance frames to better maintain the integrity of dance motions but the annotation process is time-consuming and relies on human expertise. To synthesize dance units automatically, Siyao et al. [45, 46] utilize VQ-VAE to discretize the dance motions and increase the action diversity and generative capacity. Later, Diffusion-based models [35, 36, 49] further elevated the diversity and controllability of dance synthesis results. Meanwhile, some other works [29, 30, 44, 52, 56] conduct some explorations on generating multi-dancer dance from music input. Compared to dance synthesis, dance camera synthesis is more challenging since the dance camera movements are human-centric and consist of complete shots and shot cuts, resulting in an intermittently continuous and discontinuous sequence. DanceFormer [33] uses keyframes and in-between curves to solve the music-to-dance problem which inspired us. However, DanceFormer simplifies the problem by synchronizing keyframes with music beats and ignoring the impact of motion history on the synthesis of later keyframes and motions. These issues make it hard to apply DanceFormer to dance camera synthesis. In general, generating camera movements from music and dance encounters more complicated issues despite the extensive effort on music-to-dance synthesis.

## 3 PROBLEM FORMULATION

For the problem of 3D dance camera synthesis, the goal is to generate camera movements from given music and dance. To follow the hierarchical dance-camera-making procedure by animators, as shown in Figure 1, we propose to formulate music-dance-to-camera synthesis as a three-stage problem. Our framework takes $T$ frames of music features $\boldsymbol{m} = \{m_1, m_2, \ldots, m_T\}$ and dance motions $\boldsymbol{p} = \{p_1, p_2, \ldots, p_T\}$ as input conditions, to generate camera pose sequence $\boldsymbol{c} = \{c_1, c_2, \ldots, c_T\}$. To elaborate further on these parameters, we follow FACT [34] to extract music features $m_t \in \mathbb{R}^{35}$ using Librosa [39]. For dance poses and camera movements, we follow DanceCamera3D [51] to use global positions of 60 human joints as $p_i \in \mathbb{R}^{60 \times 3}$, and MMD format camera representation in polar coordinates as $x_i \in \mathbb{R}^{3+3+1+1}$ including the global position of the reference point, rotation and distance of the camera relative to the reference point, and camera's field of view (FOV). The three stages of our framework are formally illustrated as follows:

- **Camera Keyframe Detection Stage**: Given music and dance, we intend to generate $\boldsymbol{k} = \{k_1, k_2, \ldots, k_T\}$ from $\boldsymbol{m}, \boldsymbol{p}$, where $k_t$ being 1 and 0 respectively indicate whether the frame of the moment $t$ is a camera keyframe on the timeline.
- **Camera Keyframe Synthesis Stage**: Having obtained temporal keyframe tags $\boldsymbol{k}$, in this stage, we aim to learn a mapping from $\boldsymbol{m}, \boldsymbol{p}$ and $\boldsymbol{k}$ to camera keyframe motions $\boldsymbol{c_k} = \{c_{t_1}, c_{t_2}, \ldots, c_{t_j}, \ldots\}$, where $k_{t_j} = 1$.

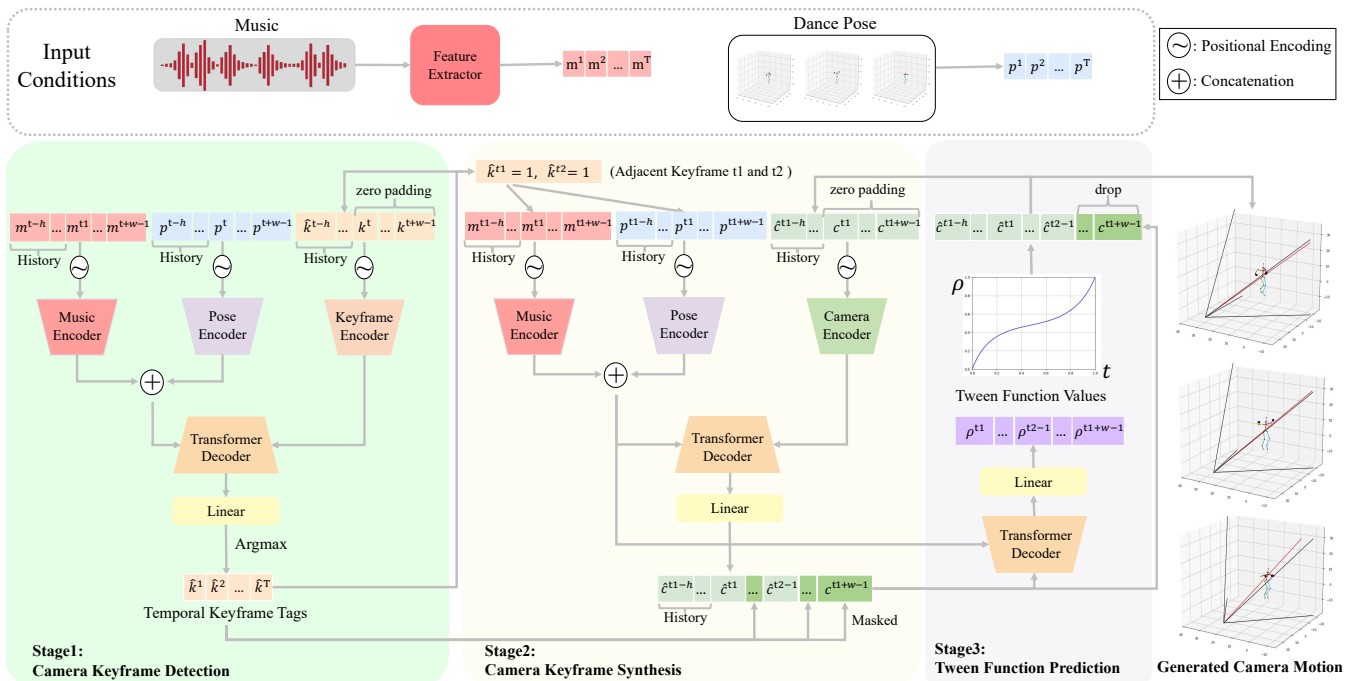

**Figure 3: Overall framework of DanceCamAnimator. In the Camera Keyframe Detection stage, the model utilizes music-dance context and temporal keyframe history to generate subsequent temporal keyframe tags. Next, for each pair of adjacent keyframes, the Camera Keyframe Synthesis stage takes music-dance context and camera history as input to synthesize camera keyframe motions. Given camera keyframe motions, camera history, and music-dance context, the final stage predicts tween function values to calculate in-between non-keyframe camera movements. Encoders with the same name share structures in different stages but are trained separately. Stages 2&3 are trained together and conducted alternately during inference.**

- **Tween Function Prediction Stage**: In this stage, our objective is to predict a tween function $\rho(t)$ for non-keyframes between two adjacent keyframes $c_{t_{j1}}$ and $c_{t_{j2}}$ from corresponding music, dance, camera motion history, and keyframe motions. So that we can calculate in-between non-keyframes camera motions as $c_t = c_{t_{j1}} + \rho(t)(c_{t_{j2}} - c_{t_{j1}})$. In this way, we obtain camera motions in all frames.

## 4  METHODOLOGY

As illustrated in Figure 3, we design a three-stage framework Dance-CamAnimitor to synthesize 3D dance camera movements following the formulation in Section 3. In the first stage, the Camera Keyframe Detection model generates keyframes on the timeline given music and dance. In the second and third stages, the models iteratively synthesize camera movements with keyframe intervals as the step length. In particular, the Camera Keyframe Synthesis model produces keyframe camera motion from camera motion history and music-dance context, while the Tween Function Prediction model takes music-dance context, camera history, and camera keyframe movements as input to synthesize tween function values and computes the in-between non-keyframe camera parameters. In this way, we generate camera movements in all frames. With this design, our DanceCamAnimitor possesses keyframe-level controllability, including modifying camera keyframes' temporal positions and

spatial movements. We will further elaborate on these three stages and the keyframe-level controllability in the following subsections.

## 4.1  Camera Keyframe Detection Stage

In the animation community's dance camera-making procedure, the animators first select keyframes on the timeline when browsing the dance and music. Thus, we imitate this procedure to design a Camera Keyframe Detection stage and solve this problem in a classification manner.

Given input music and dance poses, we first extract the acoustic features $\boldsymbol{m}$ from the music following FACT [34] to use Librosa [39] and represent the dance poses with positions of 60 joints as $\boldsymbol{p}$. Then we exploit a sliding window to select music-dance context as $\boldsymbol{m}_{t-h\sim t+w-1} = \{m_{t-h}, \ldots, m_t, \ldots, m_{t+w-1}\}$ and $\boldsymbol{p}_{t-h\sim t+w-1} = \{p_{t-h}, \ldots, p_t, \ldots, p_{t+w-1}\}$, where $t$ is current frame time, $h$ is the length of reference history, $w$ is the window length. We use zeros to pad the history when predicting the initial frames. Meanwhile, we use temporal keyframe history $\{\widehat{k}_{t-h}, \ldots, \widehat{k}_{t-1}\}$ and zero padding to stitch together as $\boldsymbol{k}_{t-h\sim t+w-1} = \{\widehat{k}_{t-h}, \ldots, \widehat{k}_{t-1}, k_t, \ldots, k_{t+w-1}\}$, $k_i = 0, i \in [t, t+w-1]$. Next, we use encoders to encode the above input as $\boldsymbol{m}^{emb}_{t-h\sim t+w-1}, \boldsymbol{p}^{emb}_{t-h\sim t+w-1}$, and $\boldsymbol{k}^{emb}_{t-h\sim t+w-1}$. Using these embeddings, we employ a transformer decoder and a linear layer to obtain the probability sequence of being a keyframe as:

$$mp^{emb}_{t-h\sim t+w-1} = \text{Concat}(m^{emb}_{t-h\sim t+w-1}, p^{emb}_{t-h\sim t+w-1})$$

$$p(k_t), \dots, p(k_{t+w-1}) = \text{Linear}(\text{Decode}(mp^{emb}_{t-h\sim t+w-1}, k^{emb}_{t-h\sim t+w-1})). \tag{1}$$

Following this, we can predict whether there is a keyframe at time $t$ by comparing the probabilities as:

$$\widehat{k}_t = \text{Argmax}(p(k_t)), \tag{2}$$

where $\widehat{k}_t$ denotes the synthesized temporal keyframe tag at frame $t$. For the training of this model, we utilize the weighted binary cross-entropy loss as:

$$\mathcal{L}_{WCE} = -\frac{1}{w}\sum_{i=0}^{w-1}[\lambda*k_{t+i}*\log(p(k_{t+i}))+(1-k_{t+i})*\log(1-p(k_{t+i}))], \tag{3}$$

where $k_{t+i}$ is the ground truth temporal keyframe tag at frame $t+i$ and $\lambda$ is the weight corresponding to keyframes.

## 4.2 Camera Keyframe Synthesis Stage

After the previous stage, we have generated the temporal position of keyframes. In this stage, we intend to synthesize keyframe camera poses from the music-dance context and camera movement history. We take the camera movement history as an input condition since adjacent shots are correlated in real dance camera movements. To achieve this functionality, we designed a pattern where this stage and the next stage proceed alternately and can be trained together. In the following, we will introduce this pattern.

For each pair of adjacent keyframes at $t1$ and $t2$, we acquire the embeddings of music-dance context $mp^{emb}_{t1-h\sim t+w-1}$ as said in Section 4.1. Here we assume that $t2$ is less than $t1 + w$, and for the exceptional cases, we provide a solution and explanation in Section 5.1. Meanwhile, we fetch the synthesized camera motion $\{\widehat{c}_{t1-h}, \dots, \widehat{c}_{t1-1}\}$ as history and pad zeros for the later $w$ frames, which can be represented as $c_{t1-h\sim t1+w-1} = \{\widehat{c}_{t1-h}, \dots, \widehat{c}_{t1-1}, c_{t1}, \dots, c_{t1+w-1}\}$, $c_i = 0$, $i \in [t1, t1 + w - 1]$. Next, we use a Camera Encoder to encode camera motion condition as $c^{emb}_{t1-h\sim t1+w-1}$. With these conditions, we use a transformer decoder to generate keyframe camera poses at $t1$ and $t2 - 1$ as:

$$\widehat{c}_{t1}, \dots, \widehat{c}_{t1+w-1} = \text{Linear}(\text{Decode}(mp^{emb}_{t1-h\sim t1+w-1}, c^{emb}_{t1-h\sim t1+w-1}))$$

$$\widehat{c}_{t1}, \widehat{c}_{t2-1} = \text{Mask}((\widehat{c}_{t1}, \dots, \widehat{c}_{t1+w-1}), t1, t2 - 1), \tag{4}$$

where we use a mask to select camera poses at $t1$ and $t2 - 1$ from the generated sequence. Here we use $t2 - 1$ instead of $t2$ to avoid the reproduction of the same keyframe. This alternative will not influence the training process since we imitate the animators to synthesize monotonically increasing tween functions. To elaborate further, given any adjacent $t1'$ and $t2'$ in ground truth data, the increment from $c_{t1'}$ to $c_{t2'}$ is always positive, so that the increment from $c_{t1'}$ to $c_{t2'-1}$ is always positive.

## 4.3 Tween Function Prediction Stage

Now that we have obtained the camera poses at $t1$ and $t2 - 1$, we aim to predict the in-between camera movements by predicting tween function values $\rho(t)$ from music-dance context, camera movement history, and camera poses at $t1$ and $t2 - 1$. Intuitively, we first concatenate camera history $\{\widehat{c}_{t1-h}, \dots, \widehat{c}_{t1-1}\}$

with generated camera keyframes $\widehat{c}_{t1}$, $\widehat{c}_{t2-1}$ from the previous stage and padding zeros to get camera history-keyframe condition $\widetilde{c}_{t1-h\sim t1+w-1} = \{\widehat{c}_{t1-h}, \dots, \widehat{c}_{t1}, \dots, \widehat{c}_{t2-1}, c_{t2}, \dots, c_{t1+w-1}\}$, $c_i = 0$, $i \in [t1+1, t2-2]$ or $[t2, t1+w-1]$. Then, we encode $\widetilde{c}_{t1-h\sim t1+w-1}$ as $\widetilde{c}^{emb}_{t1-h\sim t1+w-1}$. Next, we decode the music-dance context and camera history-keyframe condition to get the tween function values $\widehat{\rho}_{t1\sim t2-1} = \{\widehat{\rho}_{t1}, \widehat{\rho}_{t1+1}, \dots, \widehat{\rho}_{t2-1}\}$ as illustrated in Algorithm 1.

---

**Algorithm 1** Generation of Tween Function Values

1: $\Delta\widetilde{\rho}_{t1\sim t1+w-1} = \text{Linear}(\text{Decode}(mp^{emb}_{t1-h\sim t1+w-1}, \widetilde{c}^{emb}_{t1-h\sim t1+w-1}))$

2: $\Delta\widetilde{\rho}_{t1\sim t2-1} = \text{Mask}(\Delta\widetilde{\rho}_{t1\sim t1+w-1}, [t1, t2-1])$
3: $\Delta\breve{\rho}_{t1\sim t2-1} = \Delta\widetilde{\rho}_{t1\sim t2-1} - \text{Min}(\Delta\widetilde{\rho}_{t1\sim t2-1})$
4: $\breve{\rho}_{t1\sim t2-1} = \text{Cumsum}(\Delta\breve{\rho}_{t1\sim t2-1})$
5: $\widehat{\rho}_{t1\sim t2-1} = \text{Normalize}(\breve{\rho}_{t1\sim t2-1})$
6: **return** $\widehat{\rho}_{t1\sim t2-1}$

---

In particular, we first utilize the transformer decoder and linear layer to produce intermediate variables $\Delta\widetilde{\rho}_{t1\sim t1+w-1}$ and use a mask to obtain only the results from $t1$ to $t2 - 1$ as $\Delta\widetilde{\rho}_{t1\sim t2-1}$. Since the Bezier Curves used in raw data are non-differentiable, we propose to directly predict the tween function values instead of the parameters of Bezier Curves. Following this design, we first process $\Delta\widetilde{\rho}_{t1\sim t2-1}$ for non-negativization to obtain $\Delta\breve{\rho}_{t1\sim t2-1}$ denoting the increment of the tween function. Then, we calculate the cumulative sum of $\Delta\breve{\rho}_{t1\sim t2-1}$ as $\breve{\rho}_{t1\sim t2-1}$ and conduct normalization to produce $\widehat{\rho}_{t1\sim t2-1}$ which are monotonically increasing value from 0 to 1. In this way, we produce the tween function values from $t1$ to $t2 - 1$ and we can compute the camera movements $\widehat{c}_{t1\sim t2-1}$ from $t1$ to $t2 - 1$ as:

$$\widehat{c}_t = \widehat{c}_{t1} + \widehat{\rho}(t)(\widehat{c}_{t2-1} - \widehat{c}_{t1}) \quad t \in [t1, t2-1], \tag{5}$$

With the above design, the models of Camera Keyframe Synthesis and Tween Function Prediction stages can be trained together. For the loss function, we follow DanceCamera3D [51] to use $\mathcal{L}_{rec}$, $\mathcal{L}_{vel}$, $\mathcal{L}_{acc}$ for physical realism and $\mathcal{L}_{ba}$ to help the model learn the relationship between human bodyparts and the camera field of view, which can be illustrated as follows:

$$\mathcal{L}_{rec} = ||\text{Mask}(c - \widehat{c}, [t1, t2-1])||_2^2$$

$$\mathcal{L}_{vel} = ||\text{Mask}(c' - \widehat{c}', [t1, t2-2])||_2^2$$

$$\mathcal{L}_{acc} = ||\text{Mask}(c'' - \widehat{c}'', [t1, t2-3])||_2^2, \tag{6}$$

$$\mathcal{L}_{ba} = ||Jm - \hat{J}m * Jm||,$$

where $c$ and $\widehat{c}$ denote ground truth and synthesized camera movements, respectively, $Jm$ and $\hat{J}m$ are the joint masks of ground truth and generated results indicating whether each joint is inside the camera view or not. To prevent abrupt shot switches from affecting the smoothness of complete shots, we use masks to select camera motions in a complete shot given each pair of adjacent keyframes at $t1$ and $t2$. Our overall training object is the weighted sum of these losses as:

$$\mathcal{L} = \lambda_{rec}\mathcal{L}_{rec} + \lambda_{vel}\mathcal{L}_{vel} + \lambda_{acc}\mathcal{L}_{acc} + \lambda_{ba}\mathcal{L}_{ba}. \tag{7}$$

| Method | Quality | | Diversity | | Dancer Fidelity | | User Study |
|---|---|---|---|---|---|---|---|
| | $\text{FID}_k \downarrow$ | $\text{FID}_s \downarrow$ | $\text{Dist}_k \uparrow$ | $\text{Dist}_s \uparrow$ | DMR$\downarrow$ | LCD$\downarrow$ | DanceCamAnimitor WinRate$\uparrow$ |
| Ground Truth | - | - | 3.275 | 1.731 | 0.00142 | - | 35.22% ± 0.71% |
| DanceCamera3D | 3.749 | 0.280 | 1.631 | **1.326** | 0.0025 | **0.147** | 83.49% ± 1.05% |
| DanceCamera3D* (Filtered) | 7.864 | 0.313 | 0.861 | 1.301 | 0.0047 | 0.151 | 64.35% ± 2.15% |
| **DanceCamAnimator** (Ours) | **3.453** | **0.268** | **3.140** | 1.293 | **0.0022** | 0.152 | - |

Table 1: Quantitative results on the DCM [51] dataset. * means we filter the results of DanceCamera3D [51] using the officially recommended denoiser and filter. - denotes that the self-comparison is meaningless.

## 4.4 Keyframe-level Controllability

Using our novel three-stage framework, the trained models possess keyframe-level controllability including modifying keyframe temporal positions and keyframe camera poses. First, the Camera Keyframe Detection model can utilize a user-designed temporal keyframe sequence as history to detect the later keyframe temporal positions. In addition, users can replace the first stage with a pre-designed temporal keyframe sequence to generate camera movements in the following 2 stages. Moreover, the users can modify the synthesized camera keyframe poses from the Camera Keyframe Synthesis stage and employ the Tween Function Prediction stage to compute the in-between non-keyframes. This also indicates that the users can isolate the third stage from the framework to synthesize dance camera movements using their pre-designed camera keyframe poses. In Section 5.3, we provide more evidence to demonstrate the keyframe-level controllability of our framework.

## 5 EXPERIMENTS

### 5.1 Experiment Setup

**Dateset** In this work, we use DCM [51], a dataset consisting of 108 pieces of animator-designed paired dance-camera-music data including camera keyframe information. To ensure the fairness of the experiment, we re-use the train and test splits provided by the original dataset, in which the length of split sub-sequences ranges from 17 to 35 seconds and the FPS is 30. For the training of our framework, in the training set, we stitch the data pieces that are adjacent in the original data so that we acquire more training data with history.

**Implementation Details** Our final models have 38.5M parameters for the first stage and 63.7M parameters for the second and third stages. We trained our models on 4 NVIDIA 3090 GPUS with a batch size of 512, and we took 5 hours to train the first stage for 3000 epochs and 17 hours to train the second and third stages for 3000 epochs. For the extraction of acoustic features, we follow FACT [34] to use Librosa [39] instead of Jukebox [12] used in DanceCamera3D [51] because Jukebox and is an autoregressive transformer based framework and has a limit to max context window, which makes it difficult to extract features of music beyond the length limit so that we can not extract all features in advance. In addition, the extraction of the Jukebox consumes much more time if we try to extract acoustic features for music between newly synthesized keyframes in our framework. Moreover, the features extracted from Jukebox have 4800 dimensions which need more space to save and will increase the size of the models. For the history length $h$

and the window length $w$ in our models as mentioned in Section 4.1, we set them to be 60 because we find 95.9% keyframe intervals in raw data are not longer than 60. For the keyframe intervals over 60 frames, we add new keyframes in between to split them with a stride of 60. This operation will not affect the monotonicity of the tween functions since any subinterval of a monotonic function remains monotonic.

### 5.2 Comparison to DanceCamera3D

**Baseline** We compare our method with the existing advanced framework DanceCamera3D [51] on the DCM dataset. We reproduce the DanceCamera3D from the source code and filter the synthesized results utilizing the officially recommended denoiser and filter as DanceCamera3D (Filtered).

**Metrics** Following DanceCamera3D [51], we evaluate the generated results from three perspectives, including quality, diversity, and dancer fidelity. For quality evaluation, we calculate the Frechet Inception Distance (FID) [17] between generated results and test set camera sequences for the shot features and kinetic features as $\text{FID}_s$ and $\text{FID}_k$. To evaluate the diversity of camera movements, we compute the average Euclidean distance (Dist) within the shot feature space and the kinetic feature space as $\text{Dist}_s$ and $\text{Dist}_k$. For the evaluation of dancer fidelity, we employ Dancer Missing Rate (DMR) and Limbs Capture Difference (LCD) from [51] to respectively calculate the ratio of frames that the dancer is outside camera view and the difference of camera captured body parts between the synthesized results and ground truth.

**Quantitative Results** As shown in Table 1, our DanceCamAnimator beats the baseline methods on the $\text{FID}_k$, $\text{FID}_s$ and DMR, and significantly outperforms the baseline methods on the $\text{Dist}_k$. For the other two metrics, although DanceCamera3D achieves higher $\text{Dist}_s$ and lower LCD, it fails to provide smooth camera movements. After the filtering process, our method achieves a performance very close to the filtered DanceCamera3D. Meanwhile, after the filtering, the $\text{FID}_k$ of DanceCamera3D increases from 3.749 to 7.864, and $\text{Dist}_k$ drops from 1.631 to 0.861, which indicates the smooth post-process will greatly influence the kinetic quality and diversity. This phenomenon also suggests that the jittering helps DanceCamera3D to obtain results closer to the ground truth distribution, but it leads to undesirable and unrealistic visual effects. In contrast, our DanceCamAnimitor achieves much better $\text{FID}_k$ and $\text{Dist}_k$ without jittering. The above results all demonstrate the effectiveness of our DanceCamAnimitor.

**User Study** To further evaluate the real visual performance of our DanceCamAnimitor, we conduct a user study among the generated

**Figure 4: Visualization Comparison. We rendered the ground truth data and results generated from our method and the baselines given a 2-second music-dance condition. Compared to the baselines, our DanceCamAnimator synthesizes dance camera movements with more shot changes in a short period of time. This comparison also shows the usage of filters in the baseline DanceCamera3D is unstable and carries the risk of erroneous smoothing, causing the character to deviate from the center of the camera view, thus validating that our designed no post-processing framework is meaningful.**

results of our methods, baseline methods, and the ground truth. Firstly, we randomly sample 10 dance-camera-music pieces from the test set. Next, for each piece, we use the music and dance to generate camera results and render dance videos with our method and baselines. Next, we combine each dance video of our method with related baseline videos and the ground truth video so that we acquire 30 video pairs. We invited 23 participants to view these 30 video pairs in random order and distinguish which camera movements better match the music and make the dance more expressive. Results are shown in Table 1. The outcomes revealed that our method outperforms DanceCamera3D by an 83.49% win rate mainly because the participants find noticeable shaking in the dance videos. Compared to the filtered DanceCamera3D, our method wins in 64.35% cases with the remaining situations attributed to some users preferring single-camera tracking dance videos. Furthermore, although the ground truth data has many well-designed camera shots, our method beats the ground truth in 35.22% of the cases where users find it hard to distinguish between real data and generated results, denoting that our method generates fluent camera movements and various shot switches similar to the ground truth.

**Case Study** We provide evidence from the perspectives of camera curve comparison and visualization to demonstrate that our method better solves the contradiction between smooth complete shots and abrupt shot switches, and achieves better visual performance.

As shown in Figure 5, given the same music-dance condition, we plot the camera curves of the ground truth data and results generated by baseline methods and our method. Compared to DanceCamera3D, our DanceCamAnimitor produces sharper transitions reflecting shot switches and smoother camera movements within each complete shot. The filtered DanceCamera3D provides smoother camera movements but fails to synthesize abrupt changes, resulting in a lack of visual experience in multi-camera angle transitions.

As shown in Figure 4, given the same 2 seconds of music and dance data from the test set, we visualize the ground truth and the generated results of our DanceCamAnimitor and baselines. Compared to baseline methods, our method synthesizes more shot changes including distance and angle towards the dancer in such a short time. Furthermore, comparing the DanceCamera3D results with and without filtering operations, we find that the usage of the filter is unstable which may lead to excessive smoothing and cause the dancer to move away from the center of the camera view or even disappear from the camera view because the filtering is conducted only on the camera movements without awareness of the dancer. The comparison of curve results also confirms this over-smoothing phenomenon, as shown in Figure 5 that the local extremes of the curves are altered after the filtering process. For example, the local maximum values of camera FOV between frames 0 and 100 underwent considerable changes after smoothing. This significant issue once again highlights the superiority of our DanceCamAnimitor framework which requires no post-processing to synthesize smooth dance camera movements.

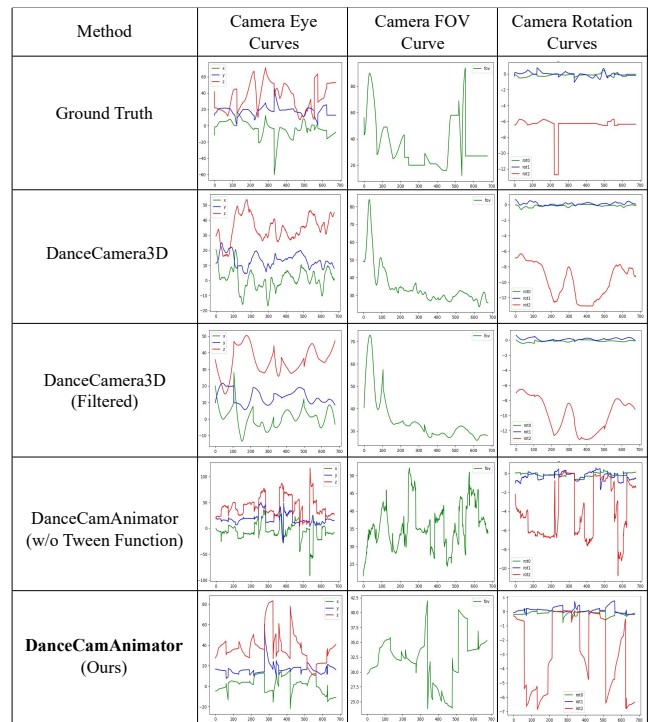

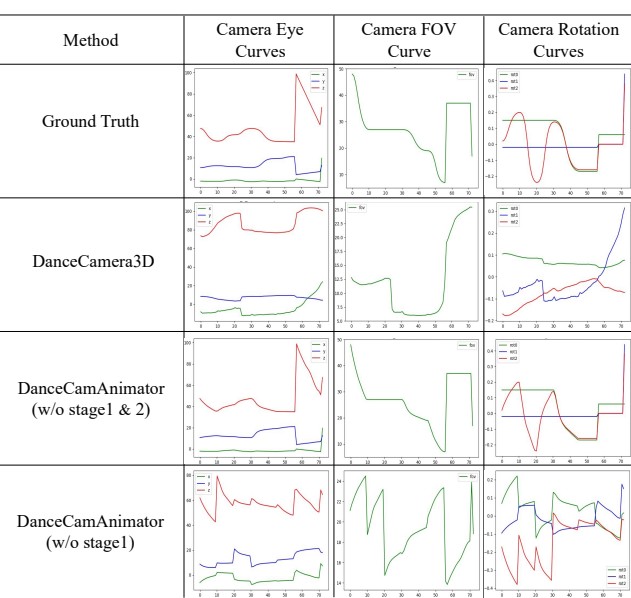

**Figure 5: Curves Comparison of Camera Parameters. Given the same music and dance input, we plot the camera curves of the ground truth and synthesized results of DanceCamera3D [51] and our DanceCamAnimator. Compared to Dance-Camera3D, our method provides more stable movements during each complete shot. Meanwhile, our method better preserves the abrupt changes caused by shot switches. If we ablate the prediction of the tween function values and directly generate camera movements, the model would fail to produce smooth shots. This demonstrates the efficacy of our design in predicting tween function values. Here camera eye represents the position of the camera in the cartesian coordinate system.**

## 5.3 Comparison on Keyframe-Level Controllability

For the comparison of keyframe-level controllability, we employ a diffusion-based editing operation in EDGE [49] on the reproduced diffusion-based DanceCamera3D as the baseline method. As shown in Figure 6, we implement keyframe temporal position control and keyframe control by replacing stage1, and stage1 & 2 with the ground truth, respectively. Compared to the ground truth, our method with keyframe control completely keeps the keyframes unchanged and synthesizes smooth in-between transitions while DanceCamera3D fails to preserve the keyframes because diffusion-based editing operation has a trade-off between keeping conditions and generating seamless transitions. Meanwhile, given only the temporal keyframe positions, our framework also synthesizes satisfying results with fluent complete shots and abrupt shot switches leveraging the stage2 & 3 models. The above results all showcase the keyframe-level controllability of our framework.

**Figure 6: Comparison on Keyframe-level Controllability. Given keyframe camera poses, our method utilizes the third stage to predict the in-between camera movements while maintaining the keyframes unchanged. In contrast, the diffusion-based method DanceCamera3D [51] fails to preserve the keyframe poses. Here we employ editing operation from EDGE [49] on the DanceCamera3D to test the controllability of the diffusion model. Besides, we use temporal keyframe positions instead of the first stage of our model at the bottom line to show the keyframe positions' controllability of our model. The keyframe temporal positions given in this figure are 0, 10, 20, 30, 45, 55, 56, 70, and 71.**

## 5.4 Ablation Study

To demonstrate the efficacy of our design to predict tween function values we ablate this operation and trained a model directly synthesizing the in-between camera movements. As shown in Figure 5, DanceCamAnimitor without tween function value prediction produces more jittering compared to our original design, indicating that our design of predicting tween function values is effective.

## 6 CONCLUSION

In this paper, we introduce DanceCamAnimitor, a three-stage 3D music-dance-to-camera movement synthesis framework that integrates the dance camera-making knowledge of animators from the animation industry. To equip our model with the capability to synthesize complete camera shots with variable lengths, we propose to alternately generate keyframes and in-between transitions. With the design of predicting tween function values rather than directly producing in-between camera poses, our framework eliminates the need for unstable post-processing that requires human intervention. Extensive experiments on the DCM standard dataset demonstrate the effectiveness and keyframe-level controllability of our method. We hope DanceCamAnimator can pave a new way for controllable dance camera synthesis.

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
