# OpenReview forum: "DanceCamAnimator: Keyframe-Based Controllable 3D Dance Camera Synthesis"
_acmmm.org/ACMMM/2024/Conference — MM2024 Poster_

### Official Review · Reviewer_2pVA · 2024-05-24

**Rating:** 3
**Confidence:** 2

**Summary:**

The paper presents DanceCamAnimator, a three-stage framework for synthesizing 3D dance camera movements from music and dance. It introduces keyframe detection, synthesis, and tween function prediction to mimic human animation procedures and provide controllable camera movements.

**Strengths:**

1. Innovative integration of animation knowledge into dance camera synthesis.

2. Provides keyframe-level controllability.

3. Eliminates the need for unstable post-processing.

**Limitations:**

1. Although the paper emphasizes the importance of keyframes, there is a lack of ablation studies on camera keyframe detection or experimental studies on the accuracy of camera keyframe detection.

2.  From the video in the Supplementary Material, it appears that the DanceCamAnimator does not provide a significant performance gain over DanceCamera3D, which is also reflected in the quality and dancer fidelity metrics in the DCM dataset.

**Suitability:**

3

---

### Official Review · Reviewer_Ck22 · 2024-05-25

**Rating:** 4
**Confidence:** 2

**Summary:**

This paper propose to integrate animator dance cinematography knowledge by formulating this task as a three-stage process, design a novel end-to-end dance camera synthesis framework. Extensive experiments on the DCM dataset demonstrate that the method
surpasses previous baselines quantitatively and qualitatively.

**Strengths:**

The paper is well organized and easy to follow.

The idea of integrating animator dance cinematography knowledge by formulating this task as a three-stage process, design a novel end-to-end dance camera synthesis framework is novel and concise.

The authors conducted adequate quantitative and qualitative experiments to verify the effectiveness of proposed method.

The details of the methods and experiments are complete.

**Limitations:**

Without a background, it is unclear how to position a camera based solely on the human body. The fact that the foreground is constantly in motion adds to the complexity and potential ambiguity of the task. Nonetheless, the task is undoubtedly intriguing.

It is also worth noting that references are generally not included in the abstract.

**Suitability:**

3

---

### Official Review · Reviewer_paU3 · 2024-05-26

**Rating:** 4
**Confidence:** 2

**Summary:**

This paper synthesizes camera movement for music and dance. Using animator dance cinematography knowledge, this paper formulate the whole process into a three-stage process: keyframe detection, keyframe synthesis, and tween function prediction. The proposed method outperforms the baseline on the DCM dataset.

**Strengths:**

1. The illustration and definitions of the problem, as well as the proposed three-stage process, are clear.
2. The results outperform the baseline method. The videos in the supplementary material are impressive, clearly generating better camera movement than competing methods.

**Limitations:**

1. The motivation for each of the three stages is not well introduced. Even though this paper clearly describes the pipeline of each stages, it is unclear why each specific stage is needed. Especially in the second camera keyframe synthesis stage, why cannot directly do the tween function prediction and skip the previous synthesis stage?
2. This paper only conducts ablation using the camera eye/FOV/rotation curves (Fig. 5 and Fig. 6). Can the authors also provide quantitative results?
3. All current experiments were conducted on anime dance videos. It would be beneficial to show the generalization ability of out-of-distribution videos, like real human dance videos.
4. Following point #1, there misses an ablation study that is without the second camera keyframe synthesis stage.



Justification: I do not see major problems in this paper, but only minor problems with motivation and experiments.

**Suitability:**

3

---

### Meta-Review · Area_Chair_2y4D · 2024-06-29

**Recommendation:** Accept (Poster)
**Confidence:** 4

**Metareview:**

This paper proposed a 3-stage process to synthesize camera motion for music and dance.
Pros: the integration of animation knowledge and dance.
Cons: lack of comprehensive ablation study. The authors are encouraged to consider beyond anime dance, e.g. what about demonstrate it also works on human dance? Its performance gain over DanceCamera3D, a competing method, is not clear.